# Assessment of the Starch-Amylolytic Complex of Rye Flours by Traditional Methods and Modern One

**DOI:** 10.3390/ma14247603

**Published:** 2021-12-10

**Authors:** Sylwia Stępniewska, Grażyna Cacak-Pietrzak, Anna Szafrańska, Ewa Ostrowska-Ligęza, Dariusz Dziki

**Affiliations:** 1Department of Grain Processing and Bakery, Prof. Wacław Dąbrowski Institute of Agricultural and Food Biotechnology—State Research Institute, Rakowiecka 36 Street, 02-532 Warsaw, Poland; anna.szafranska@ibprs.pl; 2Department of Food Technology and Assessment, Division of Fruit, Vegetable and Cereal Technology, Institute of Food Sciences, Warsaw University of Life Sciences, Nowoursynowska 159C Street, 02-776 Warsaw, Poland; grazyna_cacak_pietrzak@sggw.edu.pl; 3Department of Chemistry, Institute of Food Sciences, Warsaw University of Life Sciences, Nowoursynowska 159C Street, 02-776 Warsaw, Poland; ewa_ostrowska_ligeza@sggw.edu.pl; 4Department of Thermal Technology and Food Process Engineering, University of Life Sciences in Lublin, Głęboka 31 Street, 20-612 Lublin, Poland; dariusz.dziki@up.lublin.pl

**Keywords:** rye flour and bread, falling number, amylograph test, differential scanning calorimetry, crumb hardness, correlation analysis, PCA

## Abstract

The properties of the starch-amylolytic complex of commercial low-extract rye flour were determined based on the traditional method, such as falling number and amylograph test as well as differential scanning calorimetry (DSC). The starch, pentosans and protein had a significant effect on the thermal properties of the tested rye flours. Based on the falling number, it was revealed that rye flours were characterized by medium and low alpha-amylase activity. The falling number and amylograph test are not sufficient methods to determine the suitability of currently produced rye flours for bread making. The gelatinization process of the rye flour starch could be evaluated by the DSC test, which, together with the falling number and amylograph test, may allow a better way to evaluate the usefulness of rye flours for bread making. Many significant correlations between parameters determined by DSC endotherm and quality parameters of rye bread, such as volume and crumb hardness, were reported. Breads made from flour with higher enthalpy in DSC were characterized by higher volume and softer crumb.

## 1. Introduction

Rye (*Secale cereale* L.) is a widely grown cereal in Central, Northern and Eastern Europe. Rye grain is a good source of saccharides, proteins, fat, minerals and vitamins, mainly from group B [1]. The rye flours produced in commercial mills are mainly used for the production of various assortments of wheat-rye and rye bread. Flours intended for this purpose must have the appropriate baking value, which is mostly influenced by starch, pentosans and enzymes that break down these polysaccharides, while the effect of proteins is much smaller [2].

Rye starch, like starch of other types of cereals, is composed of two fractions–amylose and amylopectin [3]. Amylose is a linear polymer that consists of 300–1000 α-D-glucose molecules linked by α-1-4-glycosidic bonds. The amylopectin macromolecule has the same simple glucose chain as in amylose, to which every 30–40 glucose residues and side chains are attached by α-1-6-glycosidic bonds forming a three-dimensional tree structure [4,5]. Amylose and amylopectin are the two most important components in the starch, and the amylose/amylopectin ratio can have an impact on the physicochemical behavior and functionality of starch, such as the swelling, solubility, gelatinization temperature, viscosity, gelation and retrogradation properties [6].

The baking quality of rye flour is mainly determined by evaluating the starch-amylolytic complex. This complex is determined based on starch content and its properties, such as the ability to swell, gelatinization and susceptibility to the action of amylolytic enzymes. The traditional and most popular methods used for analyzing the properties of this complex are the falling number method (FN) and the amylograph test. These two tests allow us to estimate indirectly the alpha-amylase activity and flour properties related to starch susceptibility to swell and gelatinization. FN results are recorded as an index of enzyme activity in flour, and the results are expressed in seconds. In respect of rye flour, a high falling number (above 200 s) indicates low enzyme activity, while a low falling number (below 80 s) indicates high alfa-amylase activity. The amylograph analyzes viscosity by measuring the resistance of a flour and water slurry to the stirring action of pins. Generally, a thicker slurry indicates less enzyme activity. The most suitable for bread making is rye flour characterized by the FN, amylograph peak viscosity (APV) and the peak temperature of starch gelatinization (FT) in the ranges of 125–200 s [7], 400–600 AU and 63–68 °C [8], respectively. In the case of the production of rye bread from flour which is characterized by too high alpha-amylase activity (FN below 80 s, APV below 200 AU), the starch may be degraded too fast during the dough fermentation process. As a consequence, the amount of water bound by the starch is reduced, and the free water remaining in the crumb can lead to the production of bread with moist, sticky crumb, often with a protruding crust owing to overproduction of fermentation gases and collapse of bread crumb structure. Furthermore, bread produced with low alpha-amylase activity flour (FN above 200 s, APV above 700 AU) will be an unsatisfactory quality. The dough with such flour will be stiff, and the obtained bread will be low in volume, slightly acidified, little aromatic, with a compact and firm or even crumbly crumb [9,10].

An important problem in the cultivation of rye has been a tendency to sprout, mainly in years with high rainfall, in the pre-harvest period and during grain harvest. Consequently, the produced rye flours are characterized by high alpha-amylase activity. As a result of the breeding work carried out over the last decades to obtain rye varieties with increased resistance to grain sprouting and with a lower activity of α-amylase, currently, the problems with the high amylolytic activity of the rye flours are less frequent. However, it does not always guarantee good quality bread. Often, rye flours have a falling number and a final gelatinization temperature above the optimal range advised in the literature, that is, for a falling number from 125 to 200 s, and peak gelatinization temperature from 63 to 68 °C. Based on the above parameters, it is not possible to predict the suitability of flour for the production of rye bread. However, it should be emphasized that the use of flour with the optimal activity of amylolytic enzymes does not always guarantee good quality bread [11]. For this reason, it is necessary to extend the kind of tests to better assess the baking value of rye flour. One of the modern methods used to measure the thermal behavior of starches during heating is differential scanning calorimetry (DSC) [12]. During the DSC test, which is similar to the amylograph one, information about gelatinization temperatures is obtained. Additionally, the DSC test provides information about gelatinization enthalpy, which determines the amount of heat supplied during the test to equalize the temperatures of the reference sample and the tested material. This method was used by research units for evaluating the thermal properties of rye starch isolated from rye grain or rye flour [3,13]. So far, the DSC test has not been used for the evaluation of the properties of the starch-amylolytic complex of commercial rye flour.

The aim of the presents study was to evaluate the properties of the starch-amylolytic complex of rye flours based on both the traditional methods and the modern one. Furthermore, it will be important to obtain information on whether the DSC test can be used to assess the baking quality of the current production of rye flour and its suitability for bread making.

## 2. Materials and Methods

### 2.1. Materials

Ten samples of low extract rye flour (coded by letters RF1 to RF10) delivered from industrial mills situated in different regions of Poland were used in this study.

### 2.2. Methods

#### 2.2.1. Chemical Composition of Rye Flour

The following chemical components of the tested rye flour samples were analyzed: moisture content (MO), according to Standard ISO 712:2009 [14] using a conventional oven SUP 65 (WAMED, Warsaw, Poland); protein content (Pro), by Kjeldahl’s method (N·6.25) using a Kjeltec apparatus 2200 (Foss, Hillerød, Sweden) according to Standard ISO 20483:2013 [15]; ash content (AC), using an electric muffle furnace (Nabertherm, Germany, MODEL L9/R) according to Standard ISO 2171:2007 [16]; pentosans content (PC), using methodology described by Hashimoto et al. [17]; starch content (S), using polarimeters (Optical Activity, Ramsey, Cambridge shire, United Kingdom) by Ewers polarimetric method according to Standard ISO 10520:1997 [18].

#### 2.2.2. Properties of a Starch–Amylolytic Complex of Rye Flour

##### Falling Number and Amylograph Properties

The falling number test (FN) was determined using a Falling Number 1500 apparatus (Perten Instruments, Hägersten, Stockholm, Sweden) according to Standard ISO 3093:2009 [19], and the amylograph test was determined using an amylograph type 800145 with electronic temperature controller type 680026 (Brabender, Duisburg, Germany) according to Standard ISO 7973:1992 [20]. For the amylograph test, a suspension of 80 g of flour (based on 14% water content) and 450 cm^3^ distilled water was made.

##### Differential Scanning Calorimetry

Gelatinization temperatures were also measured by differential scanning calorimetry (DSC). The DSC test was performed using the TA Instrument Q 200 differential scanning calorimeter (New Castle, DE, USA). Approximately 8.0 mg of rye flour was weighed into the aluminum pans. Water was added to obtain the ratio of flour:water as 1:1. The pans with the product were hermetically sealed and kept for 24 h at room temperature. After this time, the pan with the analyzed rye flour and an identical empty pan (used as a reference sample) were placed in a calorimeter and then were heated from 20 to 110 °C at the rate of 10 °C min^−1^. Figure 1 shows an example of the DSC graph. The onset (T_o_, point A), peak (T_p_, point B) and conclusion (T_c_, point C) temperatures and the gelatinization enthalpy were estimated directly from the instrumental software. The gelatinization enthalpy was calculated per 1 g of flour. T_o_ corresponds to the amylograph pasting temperature. T_p_ corresponds to the temperature at which the conversion rate is the highest, i.e., the temperature at which the flour sample absorbs the most heat per unit of time. T_c_ corresponds to the amylograph peak temperature.

#### 2.2.3. Baking Trial

Rye breads were obtained during the laboratory baking trial. The bread variants were prepared with a one-stage process. Dough samples with a yield of 178% were prepared for each sample of rye flour. The dough was prepared with the following ingredients: 1000 g of rye flour (adjusted to the standard moisture content of 14%), 15 g of salt, 30 g of yeast, 8 cm^3^ of 88% lactic acid and 780 cm^3^ of water. The dough was prepared in the laboratory Turbo-mix-6,5 spiral mixer spiral (M and A Hommel GmbH, Wülfrath, Germany) at a low speed for 10 min. Dough temperature after mixing was in the range of 30–32 °C. The dough was placed in the fermentation cabinet (model EWPC901T, Galltec GmbH, Bondorf, Germany) with an electronic temperature sensor for 60 min. The temperature and relative humidity for the first fermentation step were 32 °C and 70–75%, respectively. After this time, the dough was divided into five pieces with a weight of 350 g each, which after manually forming, were placed into the tins and were proofed in the fermentation cabinet at 35 °C and 70–75% relative humidity until the dough surface reached optimal dough development. The optimal growth of the dough was determined experimentally. It was assumed that a piece of dough reaches its optimal growth when, when gently pressed with the tip of a finger, the surface of the dough slowly returns to its original state. The time at which the dough reaches optimal development was 15 to 17 min. The baking was performed in an oven (Piccolo, Wachtel Winkel, Germany) using steam (approximately 10 s) immediately after placing the loaves in the oven. The loaves were baked at 240 °C for 45 min. After baking, the loaves of bread were sprinkled with water, cooled and stored at room temperature in polyethylene bags. After 24 h, samples of bread were evaluated by the following parameters: specific volume of bread (BV) (in cm^3^ 100 g^−1^), determined using the millet seed displacement method [21]; crumb hardness, determined using the texture analyzer Instron 1140 (Zurich, Switzerland). The loaves were cut into slices 3 cm thick using a bread cutter (Bizerba B-100, Lublin. Poland). The crumb hardness was measured after one (H_24_) and three days (H_72_) after baking to assay the changes of crumb texture during storage of bread in PE bags at 20 °C. The bread crumb hardness corresponds to the maximum force needed to achieve 50% slice deformation. For the test, a probe with a diameter of 35 mm was used with a crosshead speed of 50 mm min^−1^. The crumb hardness was expressed in Newtons [N]. The increase of crumb hardness during storage of bread (IH) was calculated from the formula: IH = H_72_ − H_24_. Crumb moisture content (CM) was analyzed using the oven drying method. In this case, 10 g of bread crumb was dried in an oven at 130 °C for 60 min. The moisture content was expressed in % as the difference of the weight.

#### 2.2.4. Statistical Analysis

All tests were carried out at least in three replicates for all quality parameters. One-way analysis of variance (ANOVA) was performed, and the homogenous groups were determined by Tukey’s test. The tests were performed with the significance level of α = 0.05. The Pearson’s correlation coefficients between selected flour and bread parameters were calculated with a significance level of *p* < 0.01 and *p* < 0.05. In order to determine the extent to which the tested rye flours were diversified in terms of starch-amylolytic properties and determine which of the analyzed parameters had the greatest impact on this, principal component analysis (PCA) was performed. The principal component analysis (PCA) was performed on average values of each flour, which corresponded well with the analysis performed for all replicates. Data were analyzed using Statistica 13 software (TIBSO software, Palo Alto, CA, USA).

## 3. Results and Discussion

### 3.1. Basic Chemical Composition of Tested Rye Flour

The data concerning the basic components of the tested rye flour samples are presented in Table 1. Significant differences in the content of all components were observed. From a foods safety point, MO is an important parameter for the long-term storage of flour. High levels of water can allow the growth of microorganisms and is a critical factor for fungi growth and mycotoxin production [22]. Thus, low levels of MO are essential for a longer shelf-life of the product. In general, the internationally accepted maximum moisture in cereal grain, flour, pasta and other dry raw food materials is 14.0%. The samples RF2 and flour RF3 were characterized by the statistically lowest MO (12.4% and 12.8%, respectively), and reversely, the RF7 and RF8 flour samples contained the statistically highest value of MO (15.6% and 15.4%, respectively). The AC was determined in the range of 0.60% d.m. (RF6) to 0.88% d.m. (RF7 and RF8). The study revealed that flour RF6 was characterized by the lowest protein content (6.4% d.m.). Statistically, the highest Pro was found for samples RF5, RF7 and RF8 (9.0% d.m., 9.2% d.m. and 8.9% d.m., respectively). In the study conducted by Cardoso et al. [23], different kinds of refined rye flour were characterized by Pro in the range of 6.93 to 7.70% d.m. Pentosans content (PC) was found in the range from 4.9% d.m. (RF1) to 7.2% d.m. (RF8). Comparable to an earlier study [24], PC was positively correlated with Pro (r = 0.724; *p* < 0.05, Table 2). This is due to the presence of the protein and pentosans in the same anatomical part of the grain. Therefore, flour, which is characterized by higher Pro, also contains a higher PC content. In flour samples RF7 and RF8, starch formed 63.5% of the matter, whereas sample RF6 contained 71.1% of S. This component correlated negatively with Pro and PC (r = −0.886 and r = −0.913, *p* < 0.01, respectively; Table 2).

### 3.2. Properties of Starch–Amylolytic Complex

#### 3.2.1. Falling Number and Amylograph Properties

The value of FN, which is a measure of alpha-amylase activity, was in the range of 183 s (RF7) to 288 s (RF4 and RF6, respectively) (Table 3). According to the study of Michalska et al. [7], only two samples of rye flour tested in our study (RF1 and RF7) were characterized by the FN values in the optimal range of 125 to 200 s, which is optimal for the production of good quality bread. The rest of the rye flour samples were characterized by FN values above the optimal range, which may indicate lower specific bread volume as well as the undesirable drier texture of crumb [25]. In the study conducted by Michalska and Zieliński [26], flour obtained during laboratory milling of two rye cultivars were characterized by similar values of FN like in our study, while in a study by Cyran and Cygankiewicz [27], they determined the lower range of FN (from 118 to 175 s) for another pair of rye variety flours.

The amylograph peak viscosity (APV) was measured in the range of 275 AU (RF7) to 1045 AU (RF4) (Table 3). In our study, only suspensions from flour samples RF1, RF5, RF9, RF10 were characterized by the value of APV in the optimal range of 400–600 AU for the production of rye bread of expected quality. Verwimp et al. [8] stated that rye flour described by the APV exceeding 700 AU renders bread of low volume and non-typical shape. Three samples, RF2, RF4 and RF6, were characterized by APV values above 700 AU (Table 3). The listed rye flour samples also demonstrate higher values of FN in correspondence with the findings of Ponomareva et al. [28] and Stępniewska et al. [11]. The tested rye flour samples varied significantly in temperatures of starch gelatinization determined during the amylograph test. The RF2 sample was characterized by the lowest value of pasting temperature of starch gelatinization (IT) (51.5 °C), while the RF10 sample was characterized by the highest value of this parameter (54.0 °C). Differences in the IT between the studied rye flours may be the result of differences in the proportion of amylose to amylopectin [29] and differences in the molecular weight of starch [30]. According to Brites et al. [31], the size of the starch granules can have a significant impact on starch gelatinization temperature. Larger starch granules of the A fraction need more time to start pasting and usually reach higher gelatinization temperatures, compared to the smaller granules (B-fraction), because of their lower swelling capacity and lower efficiency of hydration. The peak temperature of starch gelatinization (FT) ranged from 66.5 °C (suspension from the flour RF7) to 77.0 °C (suspension from the RF4 and RF6). Only suspensions from rye flour samples RF1 and RF7 were characterized by peak temperatures in the optimal range, e.g., from 63 to 68 °C. Other rye flour samples were characterized by an FT above the optimal range.

#### 3.2.2. Differential Scanning Calorimetry

The rye flour samples were significantly different in terms of all parameters recorded from differential scanning calorimetry (DSC) endotherm (Table 4). The sample RF1 was characterized by the lowest onset temperature (T_o_) and peak temperature (T_p_) (55.9 and 60.8 °C, respectively). The sample RF8 was characterized by the highest value of T_o_ (value 59.3 °C), while the sample RF9 was characterized by the highest value of T_p_ (value 64.8 °C). According to Fredriksson et al. [32], the T_o_ is related to the amylose content in the starch granules. The study conducted by the above-mentioned authors showed significant negative correlations between T_o_ and the amylose content. In our study, the T_o_ obtained from the DSC endotherm was on average 4.8 °C higher than the IT determined during amylograph tests. One of the reasons for this situation may be the differences in the flour-water (F:W) ratio of studied samples by both the amylograph and DSC test (F:W = 1:6 and F:W =1:1, respectively). The sample RF5 was characterized by the lowest value of conclusion temperature (T_c_) (72.2 °C), while the highest T_c_ was observed for the sample RF7 (77.4 °C). According to Cornejo-Ramirez et al. [12], the T_c_ obtained from the DSC endotherm is higher for small starch granule fractions, influenced by differences in the chain length distribution of amylopectin. The study conducted by Sasaki [33] showed that the amylose content in starch has a significant influence on the T_c_. The above studies showed that there is a negative correlation between T_c_ and amylose content. This may suggest that the differences in T_c_ between the studied rye flours are related to the different proportions of amylose in the starch granules. The tested rye flour samples varied significantly in respect to the enthalpy of starch gelatinization. According to Tester and Morrison [34], the enthalpy reflects total crystallinity, i.e., the quantity and quality of crystallites in starch granules, which indicate that starches with a higher amylopectin content are characterized by a higher gelatinization enthalpy. In our study, RF5 was characterized by the significantly lowest enthalpy (2.4 J g^−1^); reversely of sample RF4 (6.5 J g^−1^). Differences in the starch gelatinization enthalpy values between the rye flour samples may result, among others, from the ratio of amylose and amylopectin [35] and different lipid content in starch [3]. Enthalpy of tested rye flour may also be influenced by the differences in the particle size distribution of starch granules. Radosta et al. [13] reported that small starch granule fractions have a lower gelatinization enthalpy than the large granule ones. The study carried out by Gudmundsson and Eliasson [36] stated a significant correlation between the enthalpy of starch gelatinization and the FN. However, such a relationship was not confirmed in our study, which may be the effect of the differences in the falling number values of the tested material. In the above-cited study, the experimental materials were rye samples with different alpha-amylase activities, while in our study, 90% of the flour samples were characterized by low alpha-amylase activity.

Our study revealed that the mainly chemical components of rye flour such as protein, starch and pentosans have a significant impact on all parameters recorded from the DSC endotherm (Table 2). Flour is a multi-component system whose thermal properties are affected by not only the starch content but also other major compounds naturally present in the flour [37], which limits water migration to starch molecules in a similar way to gluten in wheat starch. In the case of rye flour, pentosans and amylose form a film that coats the surface of the swollen granules and significantly affects the gelatinization process of rye starch. That process, even at low concentrations, limits the access of water and amylolytic enzymes to starch, delaying the gelatinization process. Slowing the starch gelatinization process leads to an increase in the size of the formed bubbles of CO_2_ because their stretching time is longer; this supports bread volume and the bread crumb softness [38].

### 3.3. Bread Quality Parameters

Figure 2 shows example images of bread crumbs. One of the most important quality parameters of bread is its volume. This parameter reflects the quality of flour and depends on the technological process used in breadmaking. Generally, at the same weight of bread, the higher loaf volume indicates higher quality because the bread dough had a greater capability of carbon dioxide production and retention during baking [39]. In our study, the bread volume (BV) was evaluated in the range 169–208 cm^3^ 100 g^−1^ for flour items RF9 and RF3, respectively (Table 5). In the earlier studies of Stępniewska et al. [11], variants of bread prepared from several samples of rye flour of high extract rate obtained by indirect methods using sourdough were characterized by BV in the range 171–249 cm^3^ 100 g^−1^.

The study revealed that the FN test is not a sufficient method for the prediction of the volume of rye bread. Among the tested flour samples (FR1 and FR7), their FN values indicate their possible use in the production of good-quality rye bread. Only bread obtained from sample RF1 was characterized by high BV (206 cm^3^ 100 g^−1^). Bread from sample FR7 had considerably lower volume (178 cm^3^ 100 g^−1^). The reason for this probably results from the differences in APV and protein content (Pro). Sample RF1 was characterized by APV in the optimal range (545 AU, Table 3), while APV for sample RF7 was below the optimal range (275 AU, Table 3). In addition, sample RF7 contained a statistically higher Pro (9.2% d.m.). According to Verwimp et al. [8], bread from rye flours, which are characterized by APV values exceeding 700 AU, have a low volume and round shape. Such a relationship was not found in our study. The breads characterized by low volume were obtained from rye flour with APV values in the range from 500 to 600 AU (Table 3). Moreover, our study shows that the bread obtained from rye flour samples with higher gelatinization enthalpy in the DSC test tended to have a higher BV. The correlation between BV and gelatinization enthalpy was significant (r = 0.883, *p* < 0.01; Table 2). No significant correlation was found between the BV and the amylograph parameters (Table 2). This indicates that the DSC test is a more accurate method to predict the bread volume from currently produced rye flours than the amylograph test.

The crumb hardness was analyzed one (H_24_) and three (H_72_) days after baking (Table 5). The H_24_ was evaluated in the range 33.4–62.0 N for flour items RF2 and RF9, respectively. In the case of H_72_, the breads from samples RF1, RF2 and RF3 were characterized by the significantly lowest values of this parameter (50.8 N, 51.0 N, 50.2 N, respectively), while for RF7 and RF9, the highest values of crumb hardness were observed (72.8 N and 75.1 N, respectively). In the study conducted by Ostasiewicz et al. [40], the crumb hardness of the rye bread assessed one day after baking was much lower (from 4.4 to 11.5 N). The reason for this is probably the differences in the levels of the alpha-amylase activity of rye flours used for bread making. Moreover, they [40] used rye flours with high alpha-amylase activity, while in our study, the research material was characterized by low alpha-amylase activity. The basic components of the tested rye flour, such as protein, pentosans and starch, had a significant impact on H_24_ and H_72_. The correlation coefficients between H_24_ and Pro, PC and S were: 0.766 (*p* < 0.01), 0.731 (*p* < 0.05) and −0.782 (*p* < 0.01), respectively, while the correlation coefficients between H_72_ and these parameters were: 0.789 (*p* < 0.01), 0.729 (*p* < 0.05) and −0.815 (*p* < 0.01) (Table 2), respectively. Our study revealed that rye breads with less hard crumb were obtained from rye flour samples with higher gelatinization enthalpy. The correlation coefficients between gelatinization enthalpy and H_24_ as well as H_72_ were statistically significant (−0.701, *p* < 0.05 and −0.803, *p* < 0.01, respectively) (Table 2). Significant correlations were also stated between T_p_ and H_24_, as well as H_72_ (r = 0.718 and 0.688, respectively; *p* < 0.05; Table 2).

The tested rye breads were significantly different in terms of increases in crumb hardness (IH) during the storage of bread (Table 5). This quality parameter of rye bread was in the range of 10.5 N (bread from RF10) to 19.6 N (bread from RF7). A significantly negative correlation was stated only between IH and IT (r = −0.734; *p* < 0.05; Table 2).

The crumb moisture content (CM) ranged from 45.4% (bread from RF1) to 47.7% (bread from RF10) (Table 5). The flour RF10 was characterized by a statistically higher CM compared to bread from rye flour samples: RF1, RF2 and RF3 (45.4%, 45.5% and 45.7%, respectively). In the study conducted by Buksa et al. [41], the crumb moisture of wholemeal rye breads was in the range of 43.0% to 48.0% and was mainly controlled by the level of fiber, including pentosans. The significant correlation between CM and PC (r = 0.632, *p* < 0.05; Table 2) was also stated in our study. However, there was no significant correlation between CM and other parameters determining the properties of the starch-amylolytic complex (Table 2).

### 3.4. Principal Component Analysis

Based on correlations among variables tested and linear combinations of them, the PCA method builds so-called latent variables (principal components, PCs), which extract as much data variability as possible. The multi-axes space of the original variables is usually altered by 3D-space with rectangular coordinates PC1-PC3, which are independent of each other. A condition of successful data transformation is explaining at least 70% of the original data by these triple of PCs. Further, principal components PC4 and PC5 are commonly omitted owing to their small contribution to the data scatter explanation. Within the present study, the closeness of quality among flour items RF1–RF10, as well as the potential alternation among quality features observed, were explored. The results of PCA showed that the first two principal components (PC1, PC2) explained 74.55% of the variation of original quality parameters (Figure 3A). The PC1 accounted for 58.19% of the data scatter with the major parameters, such as analytical: starch (S), protein (Pro), pentosans (PC), amylograph peak viscosity (APV), DSC enthalpy and bread quality; parameters: bread volume (BV), crumb hardness one (H_24_) and three (H_72_) days after baking. The S, APV, BV and gelatinization enthalpy are located on the right side of the loading plot. Meanwhile, the rest of the parameters are located on the left side of the loading plot. The PC2 explained 16.36% of the variation and was strongly positively related to falling number (FN) and amylograph pasting temperature (IT), as well as negatively related only to increased bread crumb hardness (IH). The analytical: Pro, PC; DSC: T_p_ and T_o_; and bread quality parameters: H_24_, H_72_ are closely located on the same side of the loading plot indicating that crumb of bread from rye flours, which are characterized by higher PC and Pro, have higher hardness. Whereas gelatinization enthalpy, S and BV are present closely on the opposite side of the loading plot. This indicates that flour, which is characterized by higher starch content, is also characterized by higher enthalpy.

The score plot (Figure 3B) clearly differentiated all the rye flour samples into separate right and left regions. The flour samples: RF5, RF7, RF8, RF9 and RF10 are located on the left side, while the rest of the samples are located on the right side. This indicates that samples of flour that are grouped on the right side are characterized by a relatively higher S, gelatinization enthalpy and lower Pro, AC, T_o_ and T_c_ compared to rye flour samples located on the left side. The breads obtained from flour on the right side of the graph compared to breads from flour marked on the left side are characterized by relatively higher BV and lower H_24_ and H_72_.

## 4. Conclusions

The tested ten rye flours were characterized by medium and low alpha-amylase activity. The study revealed that falling number and amylograph test are not sufficient to determine the baking quality of currently produced rye flours. Using flour with an optimal value of falling number for bread making does not guarantee high-quality bread. There were no significant relationships between traditional methods for assessing the properties of the starch-amylolytic complex of rye flour and the DSC test. The initial temperature of starch gelatinization determined during the amylograph test was, on average, about 4.8 °C higher than the onset temperature recorded from the DSC endotherm. The content of nutrients present in the flour, such as starch, protein and pentosans, have a significant impact on the thermal properties of tested rye flour. Generally, the rye flour samples with higher contents of protein and pentosans and lower starch content were characterized by higher onset and peak temperatures as well as lower gelatinization enthalpy. Our study revealed that breads from rye flour, which are characterized by gelatinization enthalpy above 4.0 J g^−1^, are characterized by relatively higher quality, e.g., higher volume and lower crumb hardness. From all study parameters used for the study, the starch-amylolytic complex of rye flour, a good indicator for the prediction of rye bread volume is gelatinization enthalpy. In respect of rye crumb bread hardness, the best indicators for prediction are DSC parameters such as conclusion temperature and gelatinization enthalpy. Therefore, the DSC test may find application in the determination of the starch-amylolytic complex of the current production of rye flours. Used together with traditional methods such as the falling number and amylograph test, they can better determine the suitability of rye flour for baking purposes.

## Figures and Tables

**Figure 1 materials-14-07603-f001:**
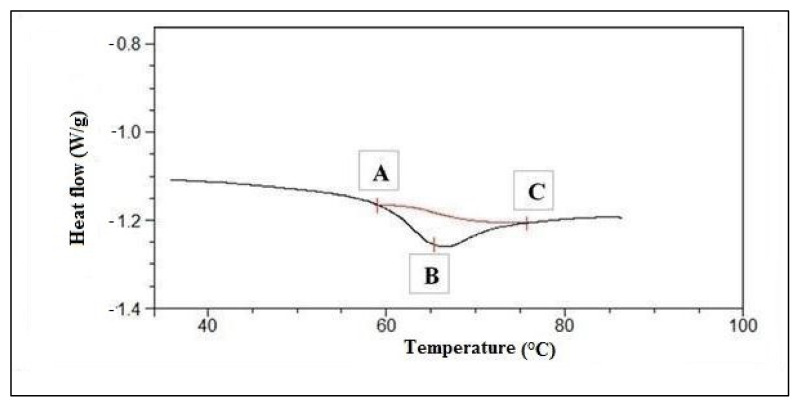
The DSC graph.

**Figure 2 materials-14-07603-f002:**
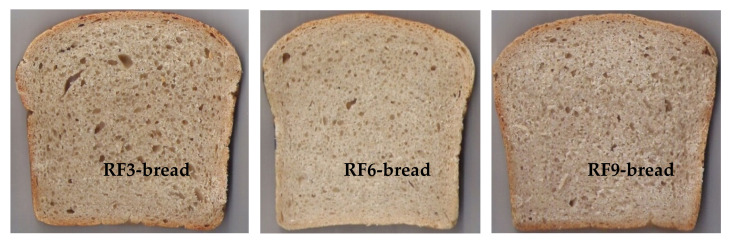
Cross-section of the obtained bread crumbs: the best (**RF3-bread**), the middle (**RF6-bread**) and the worst quality bread (**RF9-bread**).

**Figure 3 materials-14-07603-f003:**
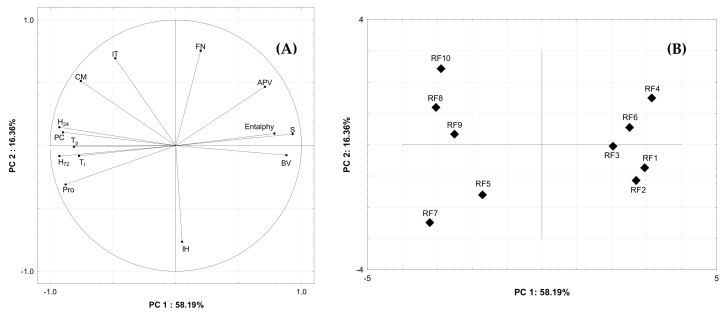
Principal component analysis: (**A**) loading plot of PC1 and PC2 for the selected analytical, amylograph, DSC and bread quality. (**B**) Score plot of PC1 and PC2 for the studied rye flours. The abbreviations are described under Table 2.

**Table 1 materials-14-07603-t001:** Chemical composition of tested rye flours.

Rye Flour	MO (%)	AC (% d.m.)	Pro (N × 6.25) (% d.m.)	PC (% d.m.)	S (% d.m.)
RF1	13.2 ± 0.07 ^c^	0.72 ± 0.02 ^a,b^	7.8 ± 0.07 ^c^	4.9 ± 0.03 ^i^	70.3 ± 0.4 ^a^
RF2	12.4 ± 0.07 ^d^	0.81 ± 0.10 ^a^	7.0 ± 0.07 ^d,e^	5.8 ± 0.04 ^f^	70.5 ± 0.6 ^a^
RF3	12.8 ± 0.14 ^d^	0.77 ± 0.04 ^a,b^	7.3 ± 0.14 ^d^	5.1 ± 0.05 ^g,h^	70.8 ± 0.3 ^a^
RF4	13.3 ± 0.00 ^c^	0.74 ± 0.01 ^a,b^	6.8 ± 0.00 ^e^	5.0 ± 0.04 ^h^	70.9 ± 0.1 ^a^
RF5	13.2 ± 0.07 ^c^	0.82 ± 0.04 ^a^	9.0 ± 0.07 ^a^	6.0 ± 0.07 ^e^	67.5 ± 0.7 ^b^
RF6	13.7 ± 0.14 ^c^	0.60 ± 0.01 ^b^	6.4 ± 0.07 ^f^	5.2 ± 0.04 ^g^	71.1 ± 0.3 ^a^
RF7	15.6 ± 0.07 ^a^	0.88 ± 0.06 ^a^	9.2 ± 0.07 ^a^	6.8 ± 0.01 ^c^	63.5 ± 0.4 ^c^
RF8	15.4 ± 0.14 ^a^	0.88 ± 0.03 ^a^	8.9 ± 0.00 ^a^	7.2 ± 0.06 ^a^	63.5 ± 0.6 ^c^
RF9	13.7 ± 0.00 ^b^	0.79 ± 0.01 ^a,b^	8.5 ± 0.14 ^b^	6.1 ± 0.03 ^d^	67.3 ± 0.4 ^b^
RF10	13.6 ± 0.14 ^b,c^	0.81 ± 0.07 ^a^	8.1 ± 0.14 ^c^	7.1 ± 0.04 ^b^	66.4 ± 0.7 ^b^

Data with the same superscript alphabets (a–i) in columns are not significantly different (*p* < 0.05). Values are mean ± standard deviation, *n* = 3. d.m., dry mass; MO, moisture content; AC, ash content; Pro, protein content; PC, pentosans content; S, starch content.

**Table 2 materials-14-07603-t002:** Pearson’s correlations coefficients between the selected rye flours and rye bread quality parameters.

Para−Maters	PC	S	FN	APV	IT	T_o_	T_p_	Enthalpy	BV	H_24_	H_72_	IH	CM
Pro	0.724 *	−0.886 **	NS	−0.826 **	NS	0.690 *	0.651 *	−0.638 *	NS	0.766 **	0.789 **	NS	NS
PC		−0.913 **	NS	NS	NS	0.701 *	0.726 *	−0.728 *	−0.710 *	0.731 *	0.729 *	NS	0.632 *
S			NS	0.701 *	NS	−0.740 *	−0.707 *	0.678 *	0.706 *	−0.782 **	−0.815 **	NS	NS
FN				0.733 *	NS	NS	NS	NS	NS	NS	NS	NS	NS
APV					NS	NS	NS	NS	NS	NS	−0.608 *	NS	NS
IT						NS	NS	NS	NS	NS	NS	−0.734 *	NS
T_o_							0.744 *	NS	NS	NS	NS	NS	NS
T_p_								NS	NS	0.718 *	0.688 *	NS	NS
Enthalpy									0.883 **	−0.701 *	−0.803 **	NS	NS
BV										−0.857 **	−0.892 **	NS	NS
H_24_											0.954 **	NS	0.735 *
H_72_												NS	0.654 *
IH													NS

** Correlation is significant at *p* < 0.01 level. * Correlation is significant at *p* < 0.05 level. NS not significant; Pro, protein content; PC, pentosans content; S, starch content; FN, falling number; APV, amylograph peak viscosity; IT, pasting temperature of starch gelatinization; T_o_, onset temperature of starch gelatinization; T_p_, peak temperature of starch gelatinization; enthalpy, gelatinization enthalpy; BV, bread volume; H_24_, H_72;_ bread crumb hardness one and three days after baking; IH., increase of bread crumb hardness; CM, bread crumb moisture.

**Table 3 materials-14-07603-t003:** Properties of starch-amylolytic complex of tested rye flours; assessment by traditional methods (falling number and amylograph test).

Rye Flour	FN (s)	APV (AU)	IT (°C)	FT (°C)
RF1	200 ± 8 ^d,e^	545 ± 7 ^e^	53.0 ± 1.4 ^a,b^	68.0 ± 0.7 ^c,d^
RF2	232 ± 4 ^c^	830 ± 14 ^b^	51.5 ± 0.4 ^b^	71.5 ± 0.7 ^b,c^
RF3	213 ± 3 ^d^	640 ± 0 ^c^	53.0 ± 0.0 ^a,b^	70.0 ± 0.4 ^b,c^
RF4	288 ± 6 ^a^	1045 ± 7 ^a^	52.5 ± 0.7 ^a,b^	77.0 ± 1.4 ^a^
RF5	216 ± 1 ^c,d^	500 ± 14 ^f^	52.0 ± 0.4 ^a,b^	70.0 ± 0.4 ^b,c^
RF6	288 ± 4 ^a^	840 ± 14 ^b^	52.5 ± 0.7 ^a,b^	77.0 ± 0.4 ^a^
RF7	183 ± 3 ^e^	275 ± 7 ^g^	52.0 ± 0.4 ^a,b^	66.5 ± 0.7 ^d^
RF8	280 ± 6 ^a^	620 ± 14 ^c,d^	52.5 ± 0.7 ^a,b^	75.0 ± 0.4 ^a^
RF9	217 ± 3 ^c,d^	590 ± 0 ^d^	53.0 ± 0.7 ^a,b^	70.0 ± 0.0 ^b,c^
RF10	254 ± 6 ^b^	525 ± 7 ^e,f^	54.0 ± 0.4 ^a^	69.5 ± 0.7 ^b,c^

Data with the same superscript alphabets (a–g) in columns are not significantly different (*p* < 0.05). Values are mean ± standard deviation, *n* = 3. FN, falling number; APV, amylograph peak viscosity; IT, pasting temperature; FT, peak temperature.

**Table 4 materials-14-07603-t004:** Properties of starch-amylolytic complex of tested rye flour assessment by modern methods (DSC test).

Gelatinization				
Rye Flour	T_o_ (°C)	T_p_ (°C)	T_c_ (°C)	Enthalpy (J g^−1^)
RF1	55.9 ± 0.7 ^e^	60.8 ± 0.4 ^f^	73.9 ± 0.3 ^c^	6.0 ± 0.1 ^b^
RF2	56.5 ± 0.4 ^b,c,d,e^	62.8 ± 0.3 ^c,d,e^	72.6 ± 0.1 ^c,d^	4.4 ± 0.2 ^c^
RF3	58.2 ± 0.4 ^a,b,c,d^	64.0 ± 0.1 ^a,b,c^	76.6 ± 0.1 ^a,b^	6.2 ± 0.2 ^b^
RF4	56.7 ± 0.8 ^c,d,e^	62.1 ± 0.4 ^d,e,f^	75.5 ± 0.4 ^b^	6.5 ± 0.1 ^a^
RF5	58.4 ± 0.3 ^a,b,c^	63.2 ± 0.1 ^b,c,d^	72.2 ± 0.3 ^d^	2.4 ± 0.3 ^g^
RF6	56.0 ± 0.7 ^d,e^	61.3 ± 0.7 ^e,f^	73.3 ± 0.4 ^c,d^	3.6 ± 0.4 ^d^
RF7	58.8 ± 0.4 ^a,b^	64.6 ± 0.1 ^a,b^	77.4 ± 0.1 ^a^	3.5 ± 0.1 ^d^
RF8	59.3 ± 0.4 ^a^	64.5 ± 0.6 ^a,b^	75.8 ± 0.4 ^b^	2.7 ± 0.4 ^f^
RF9	57.0 ± 0.3 ^b,c,d,e^	64.8 ± 0.1 ^a^	72.6 ± 0.3 ^c,d^	3.1 ± 0.2 ^e^
RF10	58.1 ± 0.8 ^a,b,c,d,e^	63.9 ± 0.4 ^a,b,c^	73.5 ± 0.1 ^c,d^	3.4 ± 0.4 ^d^

Data with the same superscript alphabets (a–g) in columns are not significantly different (*p* < 0.05). Values are mean ± standard deviation, *n* = 3. T_o_, onset temperature of starch gelatinization; T_p_, peak temperature of starch gelatinization; T_c_, conclusion temperature of starch gelatinization.

**Table 5 materials-14-07603-t005:** Baking trial results of tested rye flour.

Rye Flour	BV (cm^3^ 100 g^−1^)	H_24_ (N)	H_72_ (N)	IH (N)	CM (%)
RF1	206 ± 10 ^a,b^	37.2 ± 1.0 ^e^	50.8 ± 0.1 ^f^	13.6 ± 0.8 ^b^	45.4 ± 0.3 ^b^
RF2	201 ± 10 ^a,b,c^	33.4 ± 0.7 ^f^	51.0 ± 0.7 ^f^	17.6 ± 0.0 ^c^	45.5 ± 0.4 ^b^
RF3	208 ± 3 ^a^	37.6 ± 0.4 ^e^	50.2 ± 0.7 ^f^	12.6 ± 0.3 ^b^	45.7 ± 0.6 ^b^
RF4	199 ± 8 ^a,b,c^	41.0 ± 0.7 ^d^	54.2 ± 1.1 ^e^	13.2 ± 0.4 ^b^	46.0 ± 0.3 ^a,b^
RF5	173 ± 3 ^c,d^	52.4 ± 0.4 ^c^	69.6 ± 0.4 ^b^	17.2 ± 0.1 ^c^	46.2 ± 0.7 ^a,b^
RF6	188 ± 4 ^a,b,c,d^	38.4 ± 0.3 ^d,e^	56.6 ± 0.4 ^d^	18.2 ± 0.1 ^c,d^	46.4 ± 0.1 ^a,b^
RF7	178 ± 11 ^b,c,d^	53.2 ± 0.3 ^c^	72.8 ± 0.4 ^a^	19.6 ± 0.1 ^d^	46.4 ± 0.3 ^a,b^
RF8	179 ± 4 ^a,b,c,d^	53.0 ± 1.1 ^c^	66.5 ± 0.7 ^c^	13.5 ± 0.4 ^b^	46.5 ± 0.6 ^a,b^
RF9	169 ± 10 ^d^	62.0 ± 0.8 ^a^	75.1 ± 0.1 ^a^	13.1 ± 0.7 ^b^	46.9 ± 0.3 ^a,b^
RF10	174 ± 3 ^c,d^	56.2 ± 0.8 ^b^	66.6 ± 0.5 ^c^	10.5 ± 0.3 ^a^	47.7 ± 0.7 ^a^

Data with the same superscript letters (a–f) in the same columns are not significantly different (*p* < 0.05). Values are mean ± standard deviation, *n* = 3. BV, bread volume; H_24_, bread crumb hardness one day after baking; H_72_, bread crumb hardness three days after baking; IH, increase of bread crumb hardness; CM, bread crumb moisture.

## Data Availability

The data presented in this study are available upon request from the first author.

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
