# Peer review of "Assessment of the Starch-Amylolytic Complex of Rye Flours by Traditional Methods and Modern One"

_materials, 2021, doi:10.3390/ma14247603_

Round 1

Reviewer 1 Report

Totally, the research has been well designed. Although, there are some points to review.

Abstract L19: remove “that has not been yet used in industry to evaluate quality of rye flour” from the text.

L 21: What is the statistical significance level? Please mention it in the text (p<0.05 or p<0.01 or…). please apply in all the text where you speak about significant or insignificant.

Line 136: should be mentioned a reference for the method. Add below reference for this section

Hadidi, M.; Majidiyan, N.; Jelyani, A.Z.; Moreno, A.; Hadian, Z.; Mousavi Khanegah, A. Alginate/Fish Gelatin-Encapsulated Lactobacillus acidophilus: A Study on Viability and Technological Quality of Bread during Baking and Storage. Foods 2021, 10, 2215. https://doi.org/10.3390/foods10092215

Line 148, 154: replace " α " to “p”

Author Response

Thank  you  for taking  your  precious  time  and  giving  the valuable comments for our manuscript.

  1. Abstract L19: remove “that has not been yet used in industry to evaluate quality of rye flour” from the text.

Response: The sentence was removed from abstract

  1. L 21: What is the statistical significance level? Please mention it in the text (p<0.05 or p<0.01 or…). please apply in all the text where you speak about significant or insignificant.

Response: The information about statistical significance level was added

  1. Line 136: should be mentioned a reference for the method. Add below reference for this section

Hadidi, M.; Majidiyan, N.; Jelyani, A.Z.; Moreno, A.; Hadian, Z.; Mousavi Khanegah, A. Alginate/Fish Gelatin-Encapsulated Lactobacillus acidophilus: A Study on Viability and Technological Quality of Bread during Baking and Storage. Foods 2021, 10, 2215. https://doi.org/10.3390/foods10092215

Response: Reference was added

Line 148, 154: replace " α " to “p”

Response: : The statement was corrected

Reviewer 2 Report

The proposed manuscript presents an interesting study on the starch and amylolytic properties of a range of different rye flours.

The study design comparing falling number and amylograph measurements with differential scanning calorimetry represents an interesting approach and has the potential to contribute valuable reference values in this field of research.

However, the overall quality of the manuscript needs to be improved significantly. Especially, the structure of the manuscript (better extraction of new findings of the study and their interpretations) and the English language need to be improved.

Abstract

Sentence 1 & 3: ‘basic’ must be replaced by ‘based’

Introduction

Lines 49-50: Please rephrase sentence - should be ‘evaluating’ instead of ‘evaluated’

Lines 49-51: The reader would benefit from a brief introduction of the term starch-amylolytic complex and what exactly the term refers to in the manuscript.

Lines 64-65: Please rephrase, it is difficult to follow.

Line 68: Word missing in sentence

Lines 69-75: It might be beneficial to elaborate why exactly FN and amylograph are not sufficient to evaluate suitability of rye flours for baking. The reader would benefit from an introduction to which flour attributes are not properly described by FN and amylograph measurements and how DSC could provide an indication with regard to these attributes.

Materials and Methods

Line 89: Tables and results below refer to flours A to J (10 samples?) instead of A to G (7 samples).

Lines 110-112: Was water/flour ratio adjusted for the moisture of the flour as often done in flour analysis? If yes, a comment pointing this out should be added.

Lines 121-123: How was the water content of the formulation chosen?

Lines 125-126: What were the conditions (temperature and relative humidity) for the first fermentation step?

Lines 126-129: How was the optimal dough development determined? What were the resulting fermentation times of the formulation with flours A to J? Why was this approach chosen over an approach with the same fermentation time for all formulations? Differences in the starch and amylolytic properties of the samples would likely affect fermentation and indirectly impact bread quality through that in addition to direct structural water holding related impacts on bread quality.

This manuscript would substantially benefit from the determination of amylase activity via enzyme kits, which is also a common tool to determine flour quality and predict its baking performance. The inclusion of these values would add another dimension to this informative but concise study.

Results and Discussion

A more detailed and conclusive discussion should be provided. I would strongly suggest separating results and discussion and to discuss all the results together taking correlation analysis and PCA into consideration. This will provide a more conclusive outcome and will make it possible to describe the relations between variables of different tests better. This section could also benefit from a table summarising 1) the predictions for the bread quality with the different flours based on FN and amylograph values; 2) Potentially predictions based on DSC; 3) Actual bread quality of baking trials in the present study.

Lines 172-173: Please rephrase (‘positively correlation’).

Lines 172-173: Explanation or/and hypothesis for correlation should be provided.

Lines 174-176: Please rephrase (‘correlation negatively’).

Lines 199-201: Please rephrase, it is difficult to follow.

Lines 201-203: Might be better to discuss aspects related to bread quality outcomes later in the bread quality section of the manuscript or in combined discussion section?

Lines 234-238: Please rephrase, it is very difficult to follow.

Line 274: It would be better to refer to ‘constituents’ (as in compounds that are naturally contained in the flour) than ‘ingredients’.

Lines 213-216: It would be very beneficial to include microscopy results or particle size distribution data (Mastersizer/Lumisizer) to support the suspected correlation with APV and multiple other variables measured in this study.

Conclusion

Please highlight better, which are the assets of DSC and what would justify the adoption of routine DSC measurements to determine rye flour quality (e.g. the fact that bread volume only correlated with the enthalpy but with no other variable of the other tests apart from compositional data).

Tables and Figures

Table 2: It might be better to mention the type of tests performed in the table caption and refer to them being the traditional tests to make it easier to follow for the reader.

Table 3: It might be better to refer to DSC and this being the ‘modern’ method in the table caption (instead of thermal properties since thermal properties as such are also determined with the traditional measurements).

Table 4: Please stick with the same number of decimal places for all crumb hardness values (mean for values after three days have only one) and their standard deviations (number of decimal places does not match between means and standard deviations of hardness after three days).

For better readability of tables and included values, it would be beneficial to indicate significant differences with superscript letters behind the standard deviation values and not behind the mean values.

Please introduce abbreviations for test variables (T0, TP, BV and such) in tables 2 to 4 already to improve structure.

General comments

The quality of the English should be improved throughout the manuscript.

Units and spacing between value and unit should be checked.

Author Response

Thank  you  for taking  your  precious  time  and  giving  the valuable comments for our manuscript.

The proposed manuscript presents an interesting study on the starch and amylolytic properties of a range of different rye flours. The study design comparing falling number and amylograph measurements with differential scanning calorimetry represents an interesting approach and has the potential to contribute valuable reference values in this field of research.

However, the overall quality of the manuscript needs to be improved significantly. Especially, the structure of the manuscript (better extraction of new findings of the study and their interpretations) and the English language need to be improved.

Abstract

  1. Sentence 1 & 3: ‘basic’ must be replaced by ‘based’

Response: The sentences 1 & 3 were corrected

  1. Introduction

Lines 49-50: Please rephrase sentence - should be ‘evaluating’ instead of ‘evaluated’

Response: The sentence was corrected

  1. Lines 49-51: The reader would benefit from a brief introduction of the term starch-amylolytic complex and what exactly the term refers to in the manuscript.

Response: The general information about starch-amylolitic complex was added to the text.

  1. Lines 64-65: Please rephrase, it is difficult to follow

Response: The statement was rephrased

  1. Line 68: Word missing in sentence

Response: The missing word was added to the text

  1. Lines 69-75: It might be beneficial to elaborate why exactly FN and amylograph are not sufficient to evaluate suitability of rye flours for baking. The reader would benefit from an introduction to which flour attributes are not properly described by FN and amylograph measurements and how DSC could provide an indication with regard to these attributes.

Response: The information was added to the text

Materials and Methods

  1. Line 89: Tables and results below refer to flours A to J (10 samples?) instead of A to G (7 samples).

Response: The information was correct in the section Materials

  1. Lines 110-112: Was water/flour ratio adjusted for the moisture of the flour as often done in flour analysis? If yes, a comment pointing this out should be added.

      Response: The ratio of water/flour wasn’t adjusted for the moisture of the flour

  1. Lines 121-123: How was the water content of the formulation chosen?

Response: The water content was in range of 7 to 9 mg and was depended on the flour weight

  1. Lines 125-126: What were the conditions (temperature and relative humidity) for the first fermentation step?

Response: The temperature for the first fermentation was 32 °C and relative humidity 70%. This information was added to the manuscript in section material and methods

  1. Lines 126-129: How was the optimal dough development determined? What were the resulting fermentation times of the formulation with flours A to J? Why was this approach chosen over an approach with the same fermentation time for all formulations? Differences in the starch and amylolytic properties of the samples would likely affect fermentation and indirectly impact bread quality through that in addition to direct structural water holding related impacts on bread quality.

Response: The optimal dough development time was determined on the basis of an evaluation of the degree of expansion of a piece of dough in the mold. The optimal growth of the dough was determined experimentally. It was assumed that a piece of dough reaches its optimal growth when, when gently pressed with the tip of a finger, the surface of the dough slowly returns to its original state. The proofing time of a piece of dough was from 15 to 17 minutes. The method of determining the final proofing time was chosen in order to eliminate the defects of the bread caused by baking the dough pieces with the wrong degree of proofing the billet. These information were added to the manuscript in section material and methods. The adequate information was included in the manuscript

  1. This manuscript would substantially benefit from the determination of amylase activity via enzyme kits, which is also a common tool to determine flour quality and predict its baking performance. The inclusion of these values would add another dimension to this informative but concise study.

Response: We agree with the Reviewer, however, this study is now complete and, unfortunately, we cannot extend its scope at this stage. We will perform such determinations in our next research on the baking value of rye

Results and Discussion

  1. A more detailed and conclusive discussion should be provided. I would strongly suggest separating results and discussion and to discuss all the results together taking correlation analysis and PCA into consideration. This will provide a more conclusive outcome and will make it possible to describe the relations between variables of different tests better. This section could also benefit from a table summarising 1) the predictions for the bread quality with the different flours based on FN and amylograph values

Response: We took the reviewer's suggestion into account. However, since the relationship between FN and amylograph parameters and bread quality parameters has not been demonstrated and the quality of rye bread cannot be predicted based on these parameters, in our opinion, it is more advantageous to present the results and their discussion together than to divide them into two separate sections. The other comments of the reviewer were taken into account in the conclusion

  1. Lines 172-173: Please rephrase (‘positively correlation’).

     Response: The statement was corrected

  1. Lines 172-173: Explanation or/and hypothesis for correlation should be provided.

     Response: Explanation was added to the text

  1. Lines 174-176: Please rephrase (‘correlation negatively’).

     Response: The statement was corrected

  1. Lines 199-201: Please rephrase, it is difficult to follow.

     Response: The sentence was rephrased

  1. Lines 201-203: Might be better to discuss aspects related to bread quality outcomes later in the bread quality section of the manuscript or in combined discussion section?

Response: Aspects related to bread quality was has been transferred to the bread quality section to the paragraph describing the volume.

  1. Lines 234-238: Please rephrase, it is very difficult to follow

     Response: The statement was correct in texts

  1. Line 274: It would be better to refer to ‘constituents’ (as in compounds that are naturally contained in the flour) than ‘ingredients’.

     Response: The note was used in the text

  1. Lines 213-216: It would be very beneficial to include microscopy results or particle size distribution data (Mastersizer/Lumisizer) to support the suspected correlation with APV and multiple other variables measured in this study.

Response: We agree with the Reviewer that including the microscopy results or particle size distribution could be very beneficial. However, we are unable to extend its scope at this stage

Conclusion

  1. Please highlight better, which are the assets of DSC and what would justify the adoption of routine DSC measurements to determine rye flour quality (e.g. the fact that bread volume only correlated with the enthalpy but with no other variable of the other tests apart from compositional data).

Response: We agree with the Reviewer, appropriate changes have been made in the manuscript

Tables and Figures

  1. Table 2: It might be better to mention the type of tests performed in the table caption and refer to them being the traditional tests to make it easier to follow for the reader.

Response: Appropriate changes have been made in the manuscript

  1. Table 3: It might be better to refer to DSC and this being the ‘modern’ method in the table caption (instead of thermal properties since thermal properties as such are also determined with the traditional measurements).

Response: Appropriate changes have been made in the manuscript

  1. Table 4: Please stick with the same number of decimal places for all crumb hardness values (mean for values after three days have only one) and their standard deviations (number of decimal places does not match between means and standard deviations of hardness after three days)

Response: We agree with the Reviewer, the table contains incorrectly copied values for individual quality parameters of bread, therefore the number of decimal places for all crumb hardness values and their standard deviations were different. Appropriate changes have been made in the manuscript

  1. For better readability of tables and included values, it would be beneficial to indicate significant differences with superscript letters behind the standard deviation values and not behind the mean values

Response: We agree with the Reviewer, appropriate changes have been made in the manuscript

  1. Please introduce abbreviations for test variables (T0, TP, BV and such) in tables 2 to 4 already to improve structure

Response: Appropriate changes have been made in the manuscript

  1. General comments

The quality of the English should be improved throughout the manuscript

Units and spacing between value and unit should be checked

Response: The English language have been corrected and the units and spacing between values were corrected. 

Reviewer 3 Report

Dear authors,

you paper deals with very interesting and technologically very important thesis, that two traditional methods for evaluation of baking quality of rye flour are insufficient and outdated for such measurement of current rye flour variants (perhaps as result of rye breeding and climate changes). A lot work on the set of ten rye flour samples was carried out, but concept of gained data presentation is not arranged well. You concentrated on concise and dense presentation of primary data in the form of tables and direct simply comparison; data contrasting by use of percentage, cross-section graphs as well as the multivariate statistical method of principal components (PCA) you've left aside. Further, you've left aside potential advantages of the differential scanning calorimetry as recommended new method as well its technical comparison with the course of the traditional method - amylograph test.
Chapter M&M could be improved, some practised procedures are not described sufficiently for easy repetition of the experiment, although a link to the proper international method is given. Here you also not mentioned some quality parameters (as, e.g., difference between two values of crumb hardness IH = H72 - H24).
Within your mspt, different levels od English could be easily distinguished; the § Introduction is written more or less correctly with some typos, but the R&D part is full of fatal mistakes and superabundant substitutions (e.g., analyses # analysed; repeating verb in the following sentences, etc.). Also, there are very long sentences, in which the readers non-skilled in cereal chemistry loose the presented idea.
Further deficiency I see in missing definition of the abbreviations for all quality characteristics measured, which have to be presented within the § M&M in advance. Also the headers of the tables should comprised just the parameters abbreviations, whilst their full names should be explained within tables' footnote. Later use of non-defined abbreviations of the quality parameters is confusing.

Within the paragraph Conclusions, finally, I agree with the presented with one exception - based on presented, potential usage of the DSC apparatus will be limited in industrial mills and bakeries from a several reasons. Besides the apparatus cost, time-spending procedure, hight technical demand on users' laboratory skill and accuracy the main disadvantage I see the main one in operating with milligrams of flour, trying to extrapolate results to tons of such food material.

Please find all my suggestion within the file attached - my final opinion is "Accept after major revision" with final check of the text by native speaker.

With regards

Reviewer

Author Response

Thank you for taking your precious time and giving the valuable comments for our manuscript. We appreciate the amount of work you put into checking our manuscript.

  1. You paper deals with very interesting and technologically very important thesis, that two traditional methods for evaluation of baking quality of rye flour are insufficient and outdated for such measurement of current rye flour variants (perhaps as result of rye breeding and climate changes). A lot work on the set of ten rye flour samples was carried out, but concept of gained data presentation is not arranged well. You concentrated on concise and dense presentation of primary data in the form of tables and direct simply comparison; data contrasting by use of percentage, cross-section graphs as well as the multivariate statistical method of principal components (PCA) you've left aside. Further, you've left aside potential advantages of the differential scanning calorimetry as recommended new method as well its technical comparison with the course of the traditional method - amylograph test.

Response: Thank you for this comment. The conditions used in the DSC method, compared to the amylographic evaluation, are more similar to the conditions used during baking. This applies to the differences in the performance of the tested material. The yield of the tested material in the amylographic test is 660%, while in the DSC test is 200%, which is much closer to the dough yield used in the production of rye bread, which is about 170-180%.

  1. Chapter M&M could be improved, some practiced procedures are not described sufficiently for easy repetition of the experiment, although a link to the proper international method is given. Here you also not mentioned some quality parameters (as, e.g., difference between two values of crumb hardness IH = H72- H24).

Response: The chapter M&M was improved.

  1. Within your mspt, different levels od English could be easily distinguished; the § Introduction is written more or less correctly with some typos, but the R&D part is full of fatal mistakes and superabundant substitutions (e.g., analyses # analysed; repeating verb in the following sentences, etc.). Also, there are very long sentences, in which the readers non-skilled in cereal chemistry loose the presented idea.

Response: The English language of the manuscript was corrected.

  1. Further deficiency I see in missing definition of the abbreviations for all quality characteristics measured, which have to be presented within the § M&M in advance.

Response: The missing definition was added to the § M&M.

  1. Also the headers of the tables should comprised just the parameters abbreviations, whilst their full names should be explained within tables' footnote. Later use of non-defined abbreviations of the quality parameters is confusing.

Response: The adequate corrections were made.

  1. Within the paragraph Conclusions, finally, I agree with the presented with one exception - based on presented, potential usage of the DSC apparatus will be limited in industrial mills and bakeries from a several reasons. Besides the apparatus cost, time-spending procedure, high technical demand on users' laboratory skill and accuracy the main disadvantage I see the main one in operating with milligrams of flour, trying to extrapolate results to tons of such food material.

 Response: Currently, one of the important departments of large industrial mills and bakeries is a laboratory, equipped with modern research and measurement equipment. In Poland, industry laboratories often have more modern equipment than scientific institutes or universities, so we believe that the DSC apparatus could be potentially used in industrial conditions (this applies to large producers). The small weight of the flour portion needed to make this determination should not be a problem, because each batch of commercial flour composed in the mixing room from passenger flours is very uniform in terms of quality.

In addition, we have corrected the manuscript according to the comments in the attached pdf file.

Round 2

Reviewer 1 Report

The authors have revised the manuscript completely.